# Our Experience with Cyst Excision and Hepaticoenterostomy for Choledocal Cyst: A Single Center Case Review of 16 Patients

**DOI:** 10.3390/medicina58030416

**Published:** 2022-03-11

**Authors:** Laura Balanescu, Andreea Moga, Radu Balanescu, Tudor Strimbu, Ancuta Cardoneanu

**Affiliations:** 1Department of Pediatric Surgery, “Grigore Alexandrescu” Clinical Emergency Hospital for Children, 011743 Bucharest, Romania; andreea.moga@ymail.com (A.M.); radu.balanescu@umfcd.ro (R.B.); tudor.strimbu@gmail.com (T.S.); cardoneanu.anca@gmail.com (A.C.); 2Department of Pediatric Surgery, “Carol Davila” University of Medicine and Pharmacy, 050474 Bucharest, Romania

**Keywords:** choledocal cyst, hepaticoduodenoanastomosis, hepaticojejunoanastomosis

## Abstract

*Background and Objectives*: Choledocal cyst is a rare congenital disease of the biliary tree defined by dilatation of the extrahepatic and/or intrahepatic biliary ducts. Untreated, it leads to complications such as cholangitis, stone formation and malignant degeneration. The standard treatment for choledocal cyst is complete excision and subsequent biliary reconstruction via hepaticojejunostomy or hepatiocoduodenostomy. *Materials and Methods*: We report our experience with 16 pediatric cases of choledocal cyst over a 10-year period. *Results*: The predominant symptoms were nausea and jaundice, both at 62.5% (*n* = 10), followed by abdominal pain at 56.3% (*n* = 9). Ultrasonography was the diagnostic method used in all patients. Computed tomography was used in 75% (*n* = 12) and magnetic resonance imaging in 25% (*n* = 4) of cases. Age at the time of intervention ranged from 2 months to 17 years with a mean of 4 years and 5 months. The open approach was used in nine patients and the laparoscopic approach was used in seven patients, with one conversion to open surgery. Complete excision of the choledocal cyst was performed in 15 cases (93.7%), and partial excision with mucosectomy was performed in one case (6.2%). Eight patients (50%) underwent hepaticoduodenostomy and eight (50%) underwent hepaticojejunostomy, out of which one was attempted laparoscopically but was converted. We had a postoperative complication rate of 12.5% (*n* = 2) represented by anastomotic leak and pancreatitis. *Conclusions*: From our experience with these cases, we concluded that a wide hepaticoduodenostomy constitutes a favorable choice over the traditional hepaticojejunostomy, being more physiological and less time consuming.

## 1. Introduction

Choledocal cyst is a congenital disease of the biliary tree defined by dilatation of the extrahepatic and/or intrahepatic biliary ducts. The most common is type I, a fusiform dilatation of the extrahepatic common bile duct [1]. The cyst needs to be surgically excised regardless of the age at diagnosis because it can lead to complications such as cholelitiasis, pancreatitis, cholangitis and malignancy of the biliary tree [2].

The standard treatment for choledocal cyst is complete excision succeeded by hepaticoenterostomy. Roux-en-Y biliary jejunostomy and biliary duodenostomy are options for biliary reconstruction after choledocal cyst excision. Conventional hepatobiliary surgeons often opt for the Roux-en-Y jejunostomy due to its long history of safeness, being the gold standard for biliary reconstruction for many years. Biliary duodenostomy has been proposed to be a more physiologic option during reconstruction [3]. We aim to report our experience with pediatric choledocal cyst over a 10-year period and describe our experience with the presentation and management of choledocal cyst and our short and intermediate outcomes of both biliary drainage procedures after choledocal cyst excision.

## 2. Patients and Methods

We performed a retrospective study including all patients below 18 years of age with a diagnosis of choledocal cyst type I and IV, according to the Todani classification, who underwent open or laparoscopic cyst excision and subsequent biliary reconstruction (Roux-en-Y hepaticojejunostomy or hepaticoduodenostomy) at our center in a period of ten years, from January 2011 to May 2021. Data were collected through retrospective review of the patients’ electronic medical records by two independent reviewers and compared for consistency. A total of 24 patients were identified out of which 7 were excluded due to incomplete data and one due to the association of biliary atresia to the choledohal cyst.

Data about gender, clinical manifestation, pre- and postoperative laboratory findings, imaging methods performed to diagnose/confirm, type of choledocal cyst as per Todani modification of the Alonso-Lej classification, cyst size, age at surgery, surgical technique and reconstruction type and short and intermediate outcomes were analyzed.

Clinical manifestations include jaundice, abdominal pain, abdominal palpable mass, nausea and vomiting, acholic stool and hypochromic urine. The children who presented with cholangitis or pancreatitis were initially treated for it, followed by surgery after several weeks of medical treatment. All patients were investigated using abdominal ultrasonography. Computed tomography or magnetic resonance imaging were used to confirm the diagnosis and to provide a better anatomical configuration of the malformation. Cyst size was measured, and presence of intrahepatic bile duct dilation was evaluated. Choledocal cysts were classified according to the Todani classification into five types. The choledocal cysts seen in the pediatric population are types I and IV. The distinction between type I and IVA is arbitrary, because the intrahepatic ducts are rarely completely normal, suggesting they are variations of the same disease (Table 1) [4].

The procedure was performed under general anesthesia with the patient in the supine position, with a tilt to the right and head elevation of the operating table. Open or laparoscopic approaches were used according to the surgeon’s preference.

For the laparoscopic approach, five 5 mm ports were placed, one supraumbilical for the 30-degree telescope and four ports, one on the anterior axillary line, used for traction on the gallbladder, the second one on the midline line, below the xiphisternum, used for liver traction and two working ports, one on the midclavicular line and the other one on the midline, between the umbilicus and the xiphisternum. A right transverse supraumbilical incision was used for the open approach.

After dissection of the gallbladder and exposure of the anatomical relation of the cystic duct to the choledocal cyst, complete excision of the choledocal cyst was completed if feasible. The extent of cyst resection is proximally up to the hilum, until the non-dilated biliary duct, and distally up to the pancreaticobiliary junction (funneling of the dilated bile duct). If posterior cyst wall dissection could not be accomplished due to portal vein adhesions, Lilly’s procedure (partial excision of the cyst with mucosectomy) was performed. In patients where hepaticoduodenostomy was the method of choice, the remaining common bile duct was anastomosed in an end-to-side manner to the kocherized duodenum, starting on the posterior wall, using delayed absorbable continuous sutures. An essential aspect was to perform the anastomosis beyond the first duodenal part, distal to the pylorus, to prevent mechanical complications with pyloric function or emptying of the stomach, as well as to accomplish continuous bile flow.

When Roux-en-Y anastomosis was chosen, the jejunum was divided 10–20 cm from the ligament of Treitz and ascended 20–40 cm to the hepatic hilum. A termino-lateral jejuno-jejunal anastomosis was performed ensuring that the Roux limb was brought up sufficiently. The biliary enteric continuity was established with a termino-terminal hepatico-jejunal anastomosis. The small bowel length increases with the child’s age. Thus, a 40 cm Roux loop is unnecessarily long in infants and small children, a redundant loop leading to complications such as intestinal occlusion, bile stasis, ascending cholangitis, stone formation and malabsorption.

After surgery, all patients were admitted in the intensive care unit. Complete blood count, liver function tests and ultrasonography of the abdomen were conducted to evaluate the postoperative status and short-term outcomes. Short term outcomes included duration of drainage, length of hospital stay and the occurrence of short-term complications such as cholangitis, pancreatitis, anastomotic leak, intestinal obstruction, ileus and wound infection. Cholangitis was considered when the levels of total bilirubin and liver enzymes were noted above the normal range and in association with the symptoms of Charcot’s triad. The condition of pancreatitis was defined by a threefold serum amylase or lipase levels increase. Patients were discharged from the hospital after full tolerance for oral intake. Pathological reports of resected specimens were reviewed to confirm the presence of a choledocal cyst and identify any signs of malignancy.

The intermediate-term follow-up of our patients was conducted by visits to our center where clinical examination, ultrasonography and liver function tests were performed. The occurrence of intermediate-term complications was evaluated, including symptoms of biliary gastritis, biliary stones and bile duct malignancy.

## 3. Results

We studied a total of 16 patients. Six (37.5%) were male and ten (64.7%) were female. The male to female ratio was 1:1.7. Clinical symptoms varied; the classical triad of jaundice, lump and pain was present in only two patients. The predominant symptoms were nausea and jaundice, both at 62.5% (*n* = 10), abdominal pain at 56.2% (*n* = 9), followed by palpable abdominal mass, acholic stool and hyperchromic urine noticed in only 12.5 % of cases (*n* = 2). Cholangitis was the initial manifestation in seven cases (43.7%), and it was defined by the association of fever, jaundice and right upper quadrant abdominal pain with high levels of total bilirubin and liver enzymes. Paraclinical tests showed elevated liver enzymes in 56.2% of patients (*n* = 9). Three patients presented with pancreatitis. Age at the time of intervention ranged from 2 months to 17 years with a mean of 4 years and 5 months. Hospital stays averaged 11 days, ranging from 9 to 37 days. The seven patients (43.7%) that were diagnosed with cholangitis underwent the procedure after an average of 32.7 days (Table 2).

As a diagnostic approach, abdominal ultrasound was performed in all patients and identified the choledocal cyst in all cases. Dilatation of intrahepatic bile ducts was noticed in 75% of cases (*n* = 12). The size of the common bile duct during ultrasound in all of these patients ranged from 14 mm to 99 mm.

Computed tomography was used in 75% (*n* = 12) and magnetic resonance imaging in 25% (*n* = 4) of cases to confirm the diagnosis and for supplementary characterization of the cyst. The most common type of choledocal cyst was type I in 87.5% (*n* = 14), out of which type IA was found in 12.5% (*n* = 2), type IB in 12.5% (*n* = 2) and type IC in 62.5% (*n* = 10). Two patients (12.5%) had type IVA choledocal cyst (Figure 1).

The type of reconstruction was primarily decided by the surgeon’s preference, and it depended on the following factors: biliary anatomy, the surgeon’s technical familiarity with the procedure and the patient’s preoperative status. Ultimately, intraoperative findings dictated surgical judgment on which type of reconstruction to use. If too much tension was encountered in the procedure of hepaticoduodenostomy, a Roux limb was the surgical option for such cases.

The open approach was used in nine (56.2%) patients and laparoscopy was used in seven (43.7%) patients as well, with one conversion to open surgery due to difficult adhesion dissection. Complete excision of the choledocal cyst was performed in 15 cases (93.7%) and partial excision with mucosectomy was performed in one case (6.2%). Eight patients (50%) underwent hepaticoduodenostomy and eight (50%) underwent hepaticojejunostomy, out of which one was attempted laparoscopically but was converted. Of the eight patients that underwent hepaticoduodenostomy six were done laparoscopically (Figure 2) while the remaining two were open procedures. Six hepaticojejunostomies were done, one laparoscopically attempted, but conversion was necessary.

The histopathological report confirmed choledocal cyst in all patients.

The abdominal drainage was removed after an average of 8.1 days ranging from 4 to 27 days.

There was one patient that required reintervention. She had anastomotic leakage and required repositioning of the drainage. It is the case of a six-year-old girl diagnosed with type IC choledocal cyst who underwent choledocal cyst excision and hepaticoduodenostomy by laparoscopic approach. Intraoperatively, a large choledocal cyst was identified, highly vascularized, with thick walls, which was adherent to the pancreas on the posterior wall. On postoperative day 5 the abdominal drainage was removed. Two days later, the patient complained of abdominal distention and pain accompanied by nausea and vomiting. Abdominal ultrasonography identified a 42 mm fluid collection in the hepatic hilum. It was supplementary characterized by computed tomography which identified a 98/102/92 mm interhepaticoduodenal mixed collection (air, fluid, blood) that compressed the adjacent structures associated with a large amount of free intraabdominal fluid. The decision of reintervention was taken. Evacuation of the abdominal fluid and abdominal drainage was performed. Postoperative evolution was favorable, with slow regression of the collection and the abdominal drainage was removed after 27 days. Six months postoperatively the patient is doing well, with no other complications.

Four patients (25%) had ileus, out of which three had hepaticojejunostomy as a method of biliary reconstruction. All of these patients presented delayed return of bowel function, abdominal distension and radiologic findings of ileus. There was one case of pancreatitis, who postoperatively presented abdominal pain and elevated pancreatic enzymes. There were no postoperative cases of cholangitis or gastrointestinal bleeding. None of the patients had wound infection.

There was one death following multiple organ system failure, caused by Clostridium difficile sepsis. It was a three-month-old boy with a history of CMV hepatitis presenting with jaundice and acholic stool. Postoperatively, a bacterial infection triggered an acute deterioration of his chronic liver failure.

The mean follow-up period was 11.3 months. None of the patients developed symptoms suggestive of biliary reflux, and therefore no further investigations were carried out, such as contrast study or endoscopy.

## 4. Discussion

Choledocal cyst is a rare condition defined as an abnormal, disproportionate dilatation of the biliary ducts [5,6]. The 2015 Diagnostic Criteria for Congenital Biliary Dilatation were formulated by the Japanese Study Group on Pancreaticobiliary Maljunction, being the first formulation of universal guidelines [7]. In those criteria, the popular term of “choledocal cyst” was replaced with “congenital biliary dilatation” (CBD). This new term was adopted to furnish a more correct expression and to provide a universally proper understanding of this pathology and thus improved treatment outcomes worldwide.

Choledocal cysts are utterly rare in Europe and the United States. They appear more frequently in Asia, where the incidental rate is as high as 1:1000 hospital admissions [8]. Choledocal cyst etiology is not well defined. The most reputable theory was formulated by Babbitt in 1969. He proposed as an explanation an abnormal junction between the biliopancreatic ducts (ABPJ), which form a common duct which allows reflux of pancreatic enzymes into the biliary tree. This leads to increased pressure with consecutive pathological conditions in the biliary tract and pancreas such as inflammation, ectasia and conclusively dilatation [9]. The APBJ was not searched for in our patients. Another ethological theory suggests the implication of the obstruction of the common bile duct, a theory that is sustained by animal models, where ligation of the common bile duct leads to a dilatation comparable to the Todani type I choledocal cyst [10]. Abnormal function and spasm of the sphincter of Oddi has been correlated with choledocal cyst [11]. One more theory is the congenital theory of Davenport and Basu, which suggests an abnormality of ganglion cells in the slender portion of the common bile duct in children with choledocal cyst. It can determine a functional obstruction and proximal dilatation of the duct, in the same way as Hirschsprung disease or achalasia of the esophagus [12]. There are a few reports of familial cases and associated anomalies [13]. Congenital anomalies that can be associated with choledocal cyst are double common bile duct, sclerosing cholangitis, hepatic fibrosis, pancreatic cyst, annular pancreas and cardiac anomalies [14]. We have not found any congenital anomalies associated with choledocal cyst in our series.

International literature reports a 3–4:1 female to male ratio. This ratio is variable in different studies, but there is always a female prevalence [15]. We found a 1.6:1 female to male ratio. More than 60% of the choledocal cysts present during the first year of life [16]. In our study, the median age at presentation was 4 years and 5 months and only 18.7% were diagnosed in the first year of life.

The clinical presentation is often vague and nonspecific. However, the diagnosis is simplified by modern imaging techniques. The typical triad of jaundice, pain and a palpable mass can be identified in approximately 20% of presentations [17]. It is more frequently seen in children than in adults, and 85% of children have at least two features of the triad at diagnosis, in comparison with only 25% of adults [18]. We found this clinical association in only two patients. Jaundice and abdominal pain are the most common symptoms [19]. We found these symptoms in 62.5% and 56.2% of cases, respectively. Other presenting forms of presentation can be cholangitis, pancreatitis and biliary peritonitis from cyst rupture. Cholangitis was the manner of presentation in seven cases (43.7%). In infancy, choledocal cysts may present with pale stool, hepatomegaly and jaundice and may be hard to differentiate from biliary atresia. Due to the widespread use of prenatal ultrasonography, many choledocal cysts are now identified in the fetus but they can also be identified by magnetic resonance imaging [20]. Prenatally diagnosed choledocal cysts have a predisposition of developing liver fibrosis and portal hypertension early after birth. Obstructive jaundice and enlarging cysts are indicators for early surgery. Since biliary sludge starts forming as soon as two weeks of life, it might be beneficial to undergo surgical treatment within two weeks of life. Nonetheless, the ideal timing of surgery, in particular in asymptomatic newborns, has not been precisely defined. We had no cases of prenatally diagnosed choledocal cyst. Cystic biliary atresia should be considered as a differential diagnosis as it can easily mimic a choledocal cyst [7].

Ultrasonography is suitable to identify choledocal cysts in most children. Ultrasonography is the best initial imaging method for evaluating the entire intrahepatic and extrahepatic biliary system and gallbladder. It identifies a choledocal cyst as a distinctive cystic or fusiform dilatation of the common bile duct, of the intrahepatic ducts or sometimes a cyst in the porta hepatis, apart from the gallbladder. It may also identify the associated complications such as cholelithiasis, cholangitis or malignancy. Ultrasonography is also helpful for follow-up and assessment of any residual biliary dilatation after choledocal cyst surgery [21].

In some cases, advanced imaging techniques are required. Computed tomography can be necessary to confirm the diagnosis and magnetic resonance imaging to obtain data about the extent of the cyst, defects within the biliary tree and presence of the anomalous junction of the pancreaticobiliary duct [22]. Magnetic resonance cholangiopancreatography (MRCP) is at this time considered the gold standard for imaging [22] (Figure 3). We used ultrasonography as the initial imagistic diagnostic method in all patients, confirmed by computed tomography in 75% of cases and supplementarily characterized by magnetic resonance imaging in 25% of cases.

In 1959, Alonso Lej et al. published the first clinical series of patients with choledocal cyst and offered the first systematic description of choledocal cysts based on the clinical and anatomic findings in 96 cases [23]. They divided choledocal cyst into three types and described the therapeutic approach for each one. Todani et al. classified choledocal cysts into five major types and several subtypes according to cyst morphologies, out of which 90–95% are type I [24]. The incidence rates reported in the literature are 50–80% type I, 2% type II, 1.4–4.5% type III, 15–35% type IV and 20% type V [25]. The incidence rates determined in our study were consistent with the literature; 87.5% had type I choledocal cyst, and 12.5% had type IV A. Cysts of the cystic duct are not included in the Todani classification, being an entity that is even rarer than choledocal cysts, with only 14 cases reported in the literature to date. Serena Serradel et al. proposed including these cysts under a new type VI category [26].

The treatment strategy varies according to cyst type. Treatment modalities have changed from Tondani (1997) to Baison (2019), but the fundamental principles have remained unchanged (Table 3) [27]. For type I cyst, cyst excision and hepaticoenterostomy is the treatment of choice, Roux-en-Y anastomosis being the gold standard. T-tube applications and sphincteroplasty are not recommended nowadays. Drainage is associated with biliary stasis, recurrent infection, pancreatitis and cholangitis [27]. For type II cyst, diverticulectomy is now routinely performed, but it used to be described only in case reports. In type III choledocal cyst, transduodenal excision and sphincteroplasty were at the experimental stage in the past, but they are endoscopically performed with ERCP today, and surgery is used only as the second choice. For type IV cysts, cyst excision and hepaticoenterostomy is recommended. For type V cyst, resection is recommended for partial disease and liver transplant for diffuse disease [28].

Excision of type I and type IVA choledocal cysts followed by either hepaticoduodenostomy or hepaticojejunostomy has been generally accepted as surgical treatment in these patients [14].

Most choledocal cysts of Todani type I or IVA can be safely managed by the laparoscopic approach, except the ones that are perforated or in case of extensive adhesions or previous biliary surgery with consecutive disturbed anatomy [29]. The first minimally invasive choledocal cyst excision with hepaticojejunostomy for reconstruction was performed in 1995 in a 6-year-old girl with type I choledocal cyst, and the first robotic case was performed in 2006 [30,31]. A systematic review and meta-analysis in 2015 by Zhen et al. compared 5611 laparoscopic choledocal cyst excisions to 5797 open procedures [32]. They demonstrated longer operative time for laparoscopic procedures but shorter hospital stays and faster recovery of bowel function. There was no overall difference between most complications. They identified lower rates of intraoperative blood transfusion and postoperative adhesive bowel obstruction in the laparoscopic group. A second meta-analysis published in the same year by Shen et al. [33] confirmed Zhen et al.’s findings and suggested that with improvement of laparoscopic techniques this may become the preferred approach. The advantages of the laparoscopic approach include small incisions with minimal scarring, a good view of the porta hepatis, portal vein and hepatic arteries due to umbilicus-to-hepatic hilum direction of the scope, easier dissection and anastomosis due to magnified view and the possibility of overall abdominal examination [34]. Robotic excision of the choledocal cysts is now being increasingly practiced at centers where robotic surgery is available, being the second most common robotic procedure after pyeloplasty [35].

There is a scarcity of papers comparing hepaticojejunostomy and hepaticoduodenostomy after excision of the choledocal cyst in a single center, the majority of which come from the Far East. There are both advantages and disadvantages for each of the techniques, rendering the choice between them still a matter of debate [3].

Hepaticojejunostomy requires completion of two anastomoses, which significantly prolongs the operative time and sacrifices the eventuality of diagnostic endoscopy or endoscopic stenting [36]. Laparoscopic hepaticojejunostomy is particularly laborious to execute, with the creation of the limb extracorporeally. The Roux-en-Y limb defunctionalizes a certain length of jejunum [37]. The more extensive bowel manipulation in patients who undergo hepaticojejunostomy could prompt longer periods of enteral feeding [38]. Numerous studies demonstrate that complications include anastomotic leak, cholangitis and fluid collection in the gallbladder fossa within the early postoperative period, whereas late complications include anastomotic stricture, biliary stone formation and cholangitis [36]. Hepaticojejunostomy advocates suggest there are less leak-associated complications on the grounds of a lower suture tension in the Roux limb, and it also diverts enteric content away from the biliary tree, thus preventing cholangitis postoperatively [39]. Some maintain that hepaticojejunostomy grants a more durable reconstruction when Yamataka et al. recommendations are followed: end-to-end anastomosis when feasible, end-to-side anastomosis when inevitable with anastomosis of the common hepatic duct near to the closed end of the biliary pouch, careful choice of the vascular supply to the limb, and individualization of the length of the Roux [40].

The advantages of hepaticoduodenostomy consist of shorter operative times because a single anastomosis is required as opposed to three in a jejunal reconstruction, faster bowel function recovery as a result less handling and shorter hospital stays [38]. Some authors suggest that hepaticoduodenostomy leads to higher rates of reflux, which in turn contribute to higher incidence of postoperative cholangitis, anastomotic stricture and infrequently carcinoma [37]. The preconditions of a safe hepaticoduodenostomy are a satisfactory duodenum mobilization so as to be able to draw the common bile duct stump for a tension-free anastomosis. Anastomosis is contraindicated in cases where tension free suture is unachievable and should be converted to hepaticojejunostomy [41]. Todani et al. has advocated for a hepaticoduodenostomy with an ample enough stoma at the hepatic hilum, considering it the most effective approach [42].

We are in favor of hepaticoduodenostomy whenever practical because it is a more physiological method compared with hepaticojejunostomy and more straightforward to execute. Regarding laparoscopy, we consider hepaticoduodenostomy better and easier to perform, complications such as adhesive bowel obstruction being less likely. Reintervention in the event of stricture or lithiasis is easier with hepaticoduodenostomy as there is no Roux limb [33]. Nonetheless, some surgeons prefer the Roux-en-Y approach on account of a lasting safety record, being considered by some authors the gold standard for biliary reconstruction [37]. Some centers described more endoscopy-confirmed bilious gastritis by bile reflux, adhesive bowel obstruction and cholangitis in the hepaticoduodenostomy group [43] and Todani et al. [44] reported a hilar bile duct carcinoma 19 years after hepaticoduodenostomy. It was considered that duodenal content reflux into the bile ducts via the hepaticoduodenal anastomosis may have been harmful to the mucosal lining. A 2013 meta-analysis comparing hepaticoduodenal with hepaticojejunal anastomoses in 679 patients indicated the techniques are comparable in terms of most outcomes, excluding greater rate of gastric reflux [37]. A 15-year review by Silva-Baez et al. found a 25% complication rate in hepaticojejunostomy compared to 16.6% in hepaticoduodenostomy [5]; complications included cholangitis, leak and postoperative reflux. There were no deaths. The authors support the use of duodenum as the safer alternative.

In our study, length of hospital stay varied greatly from 9 days to 37 days, with an average of 11 days. Other authors reported average admissions ranging between 8 and 12.2 days [4,36]. Global morbidity of surgical choledocal cyst treatment varies between 13% to 40%, most consisting of wound-related complications or bilioenteric anastomotic leaks [45]. Bile leak is also the predominant mortality cause [46].

Anastomotic leak is an early complication. One patient (6.2%) developed anastomotic leak; a rate comparable to that of the literature [47]. Our case was managed conservatively, only requiring repositioning of the drainage. However, caution should be exercised when signs of peritonitis and hemodynamic instability develop, because these warrant exploration and revision of the anastomosis.

Abdominal drainage time was analyzed and averaged 8.1 days. Important fluid discharge, both in quantity and duration, suggests lymphatic damage caused by dissection of the cyst and the perihilar region [36].

The occurrence of cholangitis after reconstruction is thought to be caused by bile stasis, as well as reflux of duodenal content into biliary tree with bacterial overgrowth and consecutive partial obstruction. A Roux-en-Y anastomosis makes use of a defunctionalized, relatively long and peristaltic limb that provides sheltering to the ducts. Hepaticoduodenostomy is hypothetically associated with higher rates of cholangitis because of the direct path of enteric flora from the duodenum into the biliary tree [37]. We had no cases of postoperative cholangitis.

None of the patients developed symptoms specific for biliary reflux gastritis; hence, they did not undergo endoscopy, nor did they develop anastomotic strictures during the follow-up period. However, the presence of duodenogastric biliary reflux is difficult to determine by symptomatology only, so specific examination techniques are needed to diagnose it. Shimotakahara et al. objectified postoperative biliary gastritis in 33% of patients that underwent a hepaticoduodenostomy and in 7% of patients that underwent hepaticojejunostomy, but interestingly enough, no difference was found in the incidence of anastomotic stricture [43]. Postoperative cholangitis incidence is well correlated with anastomotic or supra-anastomotic strictures, which seems to be linked to the surgeon’s experience rather than the method for biliary-enteric anastomosis [44]. Some authors considered the hilar anastomosis at the bifurcation preferable to a distal anastomosis for type I cysts, in order to provide a broader anastomosis. It is thought that biliary reconstruction close to the pyloric ring could impede continuous excretion of the bile into the duodenum [33]. Therefore, a crucial point in hepaticoduodenostomy is to perform it beyond the first part of the duodenum, subsequent to a proper Kocher maneuver, in order to avoid any disturbance in pyloric function or gastric evacuation [37]. Another non-neoplastic late complication that might occur is intrahepatic biliary lithiasis [48] however we could not identify any such case in our study.

Biliary malignancy incidence in patients diagnosed with choledocal cyst is twice as high as the general population. Pathogenesis for these premalignant lesions is not sufficiently explained yet. The overall risk of biliary malignancy is 10–15% in East Asian countries [28], however, these data originate from the East, but rates differ in the West. In 2019 Baison et al. compared all single and multi-center reports from the East and West and discovered a malignancy rate of 0–17% in the East and 3–8% in the West, and a recurrence rate of 0–10% in the East and 3–8% in the West. Mortality and morbidity rates, type of choledocal cyst, patient history and symptomatology were similar [28].

Malignancy risk persists even after cyst excision (0.75–5.4%). Neoplastic complications after surgically treated choledocal cysts are difficult to assess and require extensive follow-up. The literature most often reports them in association with intrahepatic duct dilation, biliary lithiasis, recurrent cholangitis and strictures [49]. Koike et al. suggest that excision extent could be correlated with neoplastic complication incidence [27]. Malignancy risk increases with each decade of unoperated life; malignancy risk is 1% at 10 years, 15% at 20 years, 26% at 40 years and 45% at 70 years. Hence, biliary cancer is rare if excision is performed at less than 10 years of age [4].

Our study has several limitations; the study was done in a retrospective manner, and we have a small number of patients and a relatively short follow-up period, standardized postoperative follow-up being imperative for long-term conclusions. We attribute the relatively short follow-up to low patient compliance and a lack of interdisciplinary follow-up protocol.

## 5. Conclusions

Choledocal cyst is a rare disease. All authors agree on the necessity of cyst excision and biliary reconstruction; however, neither technique has proven itself beneficial enough to be universally adopted yet. There are advantages and disadvantages of each technique and of each approach. Since there are no prospective randomized clinical trials, judgement and experience of the attending surgeon will determine the technique for biliary tract reconstruction.

From our experience with these patients, we concluded that a wide hepaticoduodenostomy constitutes a favorable choice instead of the traditional hepaticojejunostomy, being more physiological and less time consuming.

## Figures and Tables

**Figure 1 medicina-58-00416-f001:**
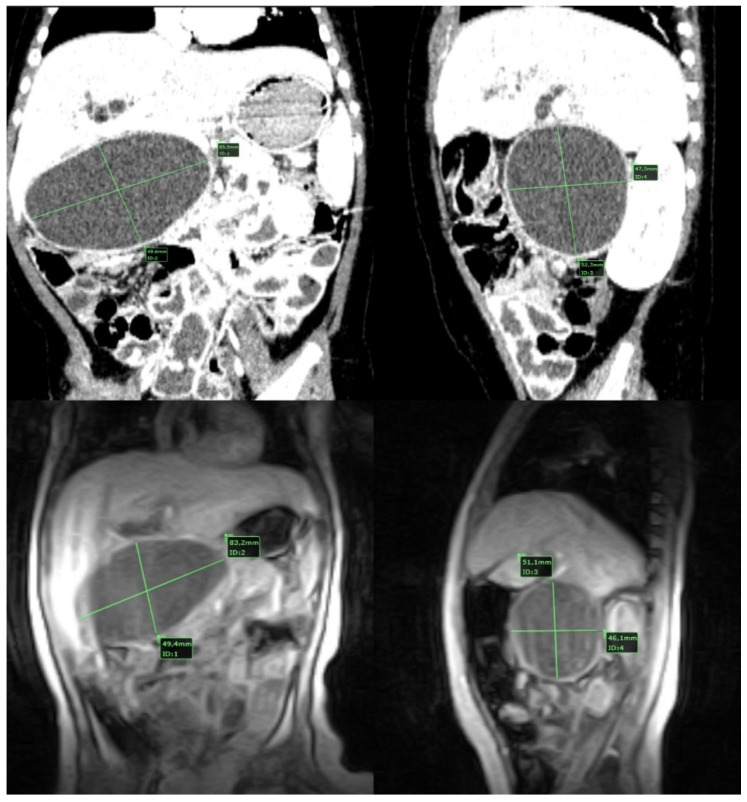
CT and MRI aspects of a type IVa large choledocal cyst.

**Figure 2 medicina-58-00416-f002:**
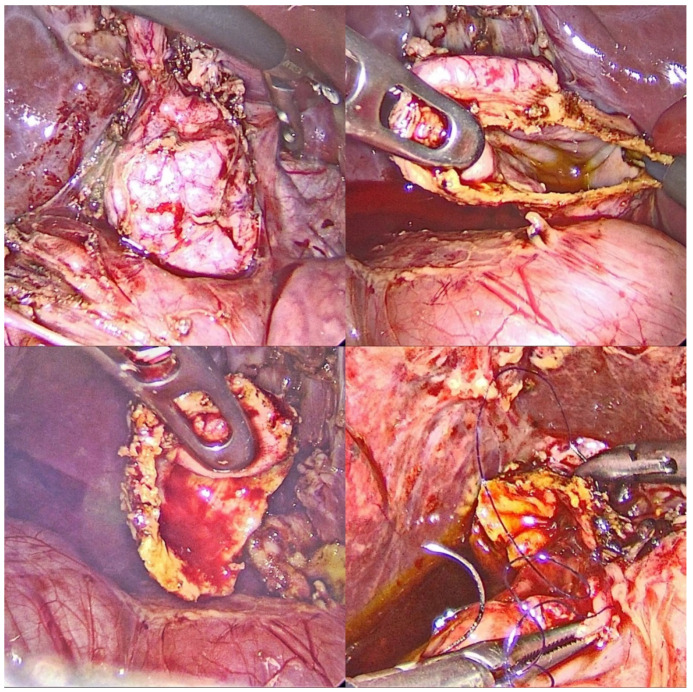
Intraoperative laparoscopic aspect.

**Figure 3 medicina-58-00416-f003:**
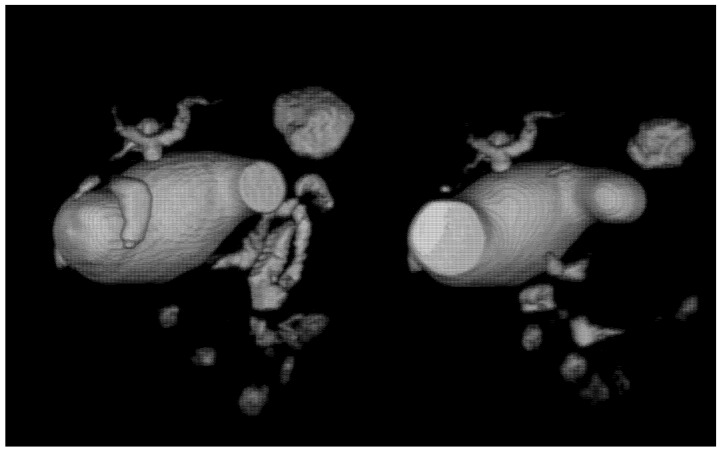
MRCP aspect of a large, type IVa choledocal cyst.

**Table 1 medicina-58-00416-t001:** Todani classification of the choledocal cyst.

Type	
IA	Diffuse cystic dilatation of the extrahepatic bile ducts, with normal intrahepatic ducts
IB	Focal, segmental cystic dilatation of the extrahepatic bile ducts
IC	Fusiform dilatation, usually extending from the pancreaticobiliary junction to the intrahepatic duct
II	A thin-stemmed diverticulum of the extrahepatic bile duct
III	Cystic dilatation of the distal extrahepatic bile duct, extending into the duodenal lumen (cholodococele)
IVA	Cystic or fusiform dilatation of the intrahepatic of extrahepatic bile ducts
IVB	Multiple cystic dilatation of the extrahepatic bile ducts (radiographically appear as a string of beads or bunch of grapes)
V	Multiple, cystic or saccular dialations of the intrahepatic bile ducts. These CCs refer to Caroli’s disease, and occur as connecting cavernous ectasia.

**Table 2 medicina-58-00416-t002:** Demographic data of the patients.

	Age (Months)	Sex	Initial Presentation							Imaging Techniques			Surgical Technique	Approach	Postop. Complications	Pathology			Follow Up (Months)
			Ns	Jd	AP	AM	As/Hu	Pt	Cl	US	CT	MRI				Cyst Type	Cyst Dimension (mm) *	Liver Disease	
1	63	F	Yes	Yes	No	No	No	No	Yes	Yes	No	No	HD	Open	No	1C	90	N	28
2	26	F	Yes	No	No	No	No	No	No	Yes	No	No	HD	Lap	No	1C	25	N	12
3	204	F	Yes	No	No	No	No	No	No	Yes	No	Yes	HD	Lap	No	1B	39	N	LTF
4	36	F	Yes	Yes	Yes	No	No	Yes	Yes	Yes	Yes	No	HJ	Open	Yes—Pt	1C	54	N	6
5	23	F	No	Yes	Yes	No	No	No	Yes	Yes	Yes	No	HD	Lap	No	4A	26	CH	12
6	120	M	Yes	No	Yes	No	No	No	No	Yes	Yes	No	HD	Lap	No	1B	30	N	48
7	35	M	Yes	Yes	Yes	No	No	No	Yes	Yes	Yes	No	HD	Lap	No	1A	42	N	16
8	18	M	No	Yes	No	No	Yes	No	No	Yes	Yes	No	HJ	Open	No	1C	48	N	10
9	14	F	No	No	Yes	No	No	No	No	yes	Yes	Yes	HJ	Open	No	4A	14	N	24
10	2	M	Yes	No	No	No	No	No	No	Yes	Yes	No	HD	Open	No	1C	30	N	36
11	82	F	Yes	Yes	Yes	Yes	No	Yes	Yes	Yes	Yes	Yes	HD	Lap	Yes—AL	1C	99	N	LTF
12	84	F	Yes	Yes	Yes	Yes	No	No	Yes	Yes	Yes	Yes	HJ	Open	No	1C	60	N	12
13	4	M	No	Yes	No	No	Yes	No	No	Yes	Yes	No	HJ	Conversion	No	1C	23	VH	Deceased
14	2	F	No	Yes	No	No	No	Yes	No	Yes	Yes	No	HJ	Open	No	1C	40	N	3
15	84	M	Yes	Yes	Yes	No	No	No	No	Yes	No	No	HJ	Open	No	1C	60	N	LTF
16	54	F	No	No	Yes	No	No	No	Yes	Yes	Yes	No	HJ	Open	No	1A	80	N	3

F = female, M = male, Ns = nausea, Jd = jaundice, AP = abdominal pain, AM = abdominal palpable mass, As/Hu = achromic stool/Hyperchromic urine, Pt = pancreatitis, Cl = cholangitis, US = ultrasonography, CT = computed tomography, MRI = magnetic resonance imaging, HD = hepaticoduodenostomy, HJ = hepaticojejunostomy, Lap = laparoscopic approach, AL = anastomotic leakage, N = no liver disease, VH = viral hepatitis, CH = cholestatic hepatitis, LTF = lost to follow-up (patient did not return for clinical and paraclinical evaluation), * Maximal diameter measured by ultrasonography.

**Table 3 medicina-58-00416-t003:** Surgical treatment from past to present.

	1977—Todani	2019—Baison
Type I	Cyst excision + Roux-en-Y HJ (T-tube for IB, sphincteroplasty for IC)	Cyst excision + Hepaticoenterostomy (HJ/HD)
Type II	No experience (Case reports − diverticulectomy)	Diverticulectomy
Type III	No experience (Case reports − transduodeal excision + sphincteroplasty)	Endoscopic transduodenal excision + sphincteroplasty
Type IV	Cyst excision + HJ	Cyst excision + Hepaticoenterostomy (HJ/HD)
Type V	Partial resection in localized disease	Partial resection for partial disease, liver transplantation for diffuse disease

## Data Availability

https://docs.google.com/spreadsheets/d/1C5xkUE1qGhUsOpXhvyb65AA_OkPelNLgzyNlIEB-GrA/edit?usp=sharing (accessed on 12 December 2021).

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
