# Peer review of "Our Experience with Cyst Excision and Hepaticoenterostomy for Choledocal Cyst: A Single Center Case Review of 16 Patients"

_medicina, 2022, doi:10.3390/medicina58030416_

Round 1

Reviewer 1 Report

  • Inference that hepaticodudenostomy is better than hepaticojejunostomy lacs evidence as no bias has been taken care of regarding selection of cases,types of choledochal cyst,less number of cases.
  • Laparoscopic management of choledochal cyst is an evolving procedure and still not a standard of care,to comment after operating six cases is premature conclusion.

Author Response

Thank you for your constructive review. Choledocal cyst is a rare occurance in pediatric care, resulting in a small number of cases per center. Therefore, we can only describe which technique had better results in our cases, without attempting  to claim a universal applicability of our findings. Because of the relatively small number of cases, cyst type could not have been selected for. The exclusion of cases was done solely due to incomplete data, with no other selection criteria.

We agree that neither of the procedures established itself as the standard of care. We state this in the conclusion where we affirm that neither technique has proven itself beneficial enough to be universally adopted yet.

Reviewer 2 Report

  • There has been reported two deaths. One secondary to Clostridium difficile, which is appropriately described. The other one due to CMV hepatitis, which is not adequately described and analyzed (rare complication after a biliary cyst excision)
  • Too short follow up period to state that both biliary reconstruction techniques (hepaticoduodenostomy vs. hepaticojejunostomy) are equal in terms of complications in the long term (although this has been stated in the discussion and conclusions)
  • The study was carried out over a period of 10 years but the mean follow up is only 11.3 months? I understand that 3 patients were lost to follow-up, and two died within the post period, but I would have expected a longer mean follow up time. Perhaps the authors can explain this.

Author Response

Thank you for your kind and constructive review.

We will provide more details on the patient with CMV infection.

We stated in the discussion that one of the limitations of the study is the relatively short follow-up.

We will provide more explanations on the possible reasons for the short follow-up period.

Reviewer 3 Report

This original paper presents a one-center retrospective study regarding choledochal cyst management in children, including a small cohort over 10 years. 

The data presented by the authors are interesting as a management experience in one surgical center from Romania, but there are issues regarding the study. Some of these issues were already underlined by the authors at the end of the paper: a small number of patients, over 10 years, small duration of the follow-up (the maximum was 4 years; so it seems challenging to note complications as malignant degeneration in such a short time). Based on this limited cohort, it is difficult to have significant conclusions on the best surgical approach (hepaticoduodenoanastomosis, or hepaticojejuno-anastomosis and open or laparoscopic surgery). 

Some improvements should be made:

  • Abstract: the way data are presented must be improved (see nausea and jaundice, both with 11 cases; a sentence should not begin with a number - line 19.
  • Patients and methods: line 52 should include the reference for Todani classification. 
  • There is no need to include table I with Todani classification but cite the paper presenting it.
  • There is no mention of the Informed consent or Ethical approval from the Hospital/University Committee of Ethics (just a standard text without any particularity for the respective medical center). This should mandatorily be included.
  • Results: presentation of the results should be improved, including editing the text (no number as the first word in a sentence; careful presentation of the data - see nausea, jaundice, abdominal pain.
  • AST and ALT should be used as abbreviations only after defining them at their first use.
  • There are no data presented regarding transaminases (just a mention, but without any evidence in the table), bilirubin level, cholestasis enzymes or pancreatic enzymes.
  • There is no need to mention the cost of hospitalization. It is not relevant (no comparison with other studies, no economic analysis presented, and more than 10 years of the study with probably many economic changes in your country).
  • Table II: all the abbreviated words should be explained in the legend.
  • As Table II already presents the cyst dimensions, there is no need for Figure 1 (moreover, there is no discussion regarding Figure 1).
  • Verify if Figure 3 is well inserted in the text: "Intraoperative laparoscopic aspect", but the anchor in the text is at the "open hepaticojejunostomies".
  • The authors should mention why they presented more in-depth those two cases with reintervention. Moreover, the authors can discuss one of these cases and explain the presence of cirrhosis at 7 months. Probably, more should be discussed on the differential diagnosis in infancy with the cystic biliary atresia. 
  • The Discussions should be improved and mainly must include discussions regarding data presented in this study corroborated with other studies. There is no need to repeat the results. MRCP is mentioned, including an image, but there is no citation - Figure 4 is from one of the patients? Also, Table III is without a citation of a reference.
  • The limitations of the study must be moved before Conclusions
  • The Conclusions should include only those supported by the data presented in this study
  • The Authors' contributions should be checked as a name is different from one from the authors' list in the manuscript.
  • The Reference list should be checked for editing - see 8, 11, 35
  • the English language used in the manuscript must be verified, if possible, by a native speaker (see "our lot" probably means our cohort or our study
  • Use the same type of numbers for the Tables in text and in the title (you have Table 1 in the text and Table I in the table's title, the same 2 and II, 3 and III).

Author Response

Thank you for your kind and constructive review. We appreciate the time and effort put into this review.

  1. We will reformulate the sentence to make both rates clear. We will also reformulate the sentence as to not begin with the number 9.
  2. The Todani classification referene will be included
  3. We will consider removing the table.
  4. We will include more details on the ethical approval
  5. We will make an overall review in order to find mistakes and to find better formulations for certain ideas.
  6. We will use the long-forms first and use the abbreviations subsequently.
  7. We did not focus on the paraclinic aspect of diagnostic and postoperative cholangitis and stasis but rather on the presence or absence of the the forementioned.
  8. We will dispose of the financial data.
  9. A legend will be added presenting the long-form of all abbreviated words in the table.
  10. We decided to remove figure 1 altogether and preserved the cyst size column in the table.
  11. We will fix the mistake in the figure description and text.
  12. We will expand on the case of the patient with hepatic cirrhosis and discuss the differential diagnosis with cystic biliary atresia.
  13. We will exclude the segments of text that repeat results in the Discussion chapter. The MRCP image is from one of our patients. A citation will be added for Table 3.
  14. We will change the order.
  15. We consider that some of the conclusions need to be mentioned even though they are not directly related to the data in the study such as the rarity of choledocal cysts and the universal acknowledgment of the necessity of a surgical sanction. However we agree that personal experience on the complication rates of the two procedures should not be mentioned and we will remove it. Study limitation will be moved as mentioned in the above point.
  16. We will remove the middle names from the Author Contributions section.
  17. We will do our best to decrease the number of English language faults.
  18. We will harmonize the number types.

We thank you again for the effort of constructing such an in-depth review of our article. We hope that our answer is satisfactory.

Reviewer 4 Report

The paper is an original study on 17 cases operated for choledochal cyst. The manuscript is well written, based on solid statistical analysis, and with rich imagistic documentation.

Minor issues:

  • some minor English spelling errors should be corrected;
  • In the discussion section, the authors could add a table to better synthesize the advantages and disadvantages of hepaticoduodenostomy and hepaticojejunostomy.

Author Response

Thank you for your kind and constructive review.

We will correct all the issues regarding grammar and spelling.

We will definitely consider adding a table to showcase advantages and disadvantages for the two types of procedures used.

Round 2

Reviewer 2 Report

  1. English must be improved
  2. The patient with cirrhosis was most probably a case of biliary atresia, and should be removed from the series in my opinion

Author Response

Thank you for your review. We excluded the patient from our study as you suggested. We also proof-read the whole text for the remaining English language errors.

Reviewer 3 Report

The authors changed the manuscript according to most of the recommendations. I do not have anything more to add as comments.

Author Response

Thank you for your review.